# Dispersion/dilution enhances phytoplankton blooms in low-nutrient waters

Yoav Lehahn[1], Ilan Koren[1], Shlomit Sharoni[1], Francesco d'Ovidio[2], Assaf Vardi[3] & Emmanuel Boss[4]

Spatial characteristics of phytoplankton blooms often reflect the horizontal transport properties of the oceanic turbulent flow in which they are embedded. Classically, bloom response to horizontal stirring is regarded in terms of generation of patchiness following large-scale bloom initiation. Here, using satellite observations from the North Pacific Subtropical Gyre and a simple ecosystem model, we show that the opposite scenario of turbulence dispersing and diluting fine-scale ($\sim$1–100 km) nutrient-enriched water patches has the critical effect of regulating the dynamics of nutrients–phytoplankton–zooplankton ecosystems and enhancing accumulation of photosynthetic biomass in low-nutrient oceanic environments. A key factor in determining ecological and biogeochemical consequences of turbulent stirring is the horizontal dilution rate, which depends on the effective eddy diffusivity and surface area of the enriched patches. Implementation of the notion of horizontal dilution rate explains quantitatively plankton response to turbulence and improves our ability to represent ecological and biogeochemical processes in oligotrophic oceans.

[1] Department of Earth and Planetary Sciences, Weizmann Institute of Science, Rehovot 76100, Israel. [2] Sorbonne Universités (UPMC, Université Paris 06)-CNRS-IRD-MNHN, LOCEAN-IPSL Laboratory, 4 Place Jussieu, 75005 Paris, France. [3] Department of Plant Sciences, Weizmann Institute of Science, Rehovot 76100, Israel. [4] School of Marine Sciences, University of Maine, Orono, Maine 04469-5741, USA. Correspondence and requests for materials should be addressed to Y.L. (email: yoav.lehahn@weizmann.ac.il) or to I.K. (email: ilan.koren@weizmann.ac.il).

Approximately half of the global primary production is carried out by oceanic phytoplankton[1]. To a first order, this immense production of biological matter is controlled by vertical motions that regulate light and nutrient availability, and stimulate the formation of phytoplankton blooms[2]. The biomass accumulated during these blooms is spatially redistributed by horizontal stirring and mixing[3–6]. As shown in a number of model studies, in addition to structuring bloom morphology, horizontal stirring may affect different aspects of bloom ecology and biogeochemistry, including regulation of plankton ecosystem dynamics and modulation of primary production levels[7–9].

Schematically, the horizontal redistribution process can be divided into two subsequent phases with contrasting effects on tracer distribution. Over short time scales, tracers are stretched and filamented by stirring, gradients are amplified and variance cascades towards smaller scales. Over longer time scales, small-scale turbulence and, ultimately, diffusion mix the tracer with its surroundings and homogenize its distribution. Classically, studies that consider the effect of stirring on phytoplankton blooms refer to the first phase of the process, emphasizing its role in generating mesoscale and sub-mesoscale ($\sim 1$–$100\,km$, hereafter referred to as fine scale) patchiness following large-scale bloom initiation[4,5,10]. The opposite scenario of fine-scale phytoplankton patches feeding variance back to scales that are larger than the size of mesoscale eddies (typically of the order of $100\,km$) through dispersion is also expected[11]. However, the process and its impact on the planktonic ecosystem have largely been overlooked.

A major reason for the limited knowledge about the effect of upscale transfer of plankton variance is the inherent difficulty to track fine-scale seeding events and to follow their evolution as they are advected and dispersed by the oceanic flow. As a result, observational evidence to the effect of horizontal stirring on localized phytoplankton blooms is limited to the case of artificial iron fertilization experiments[3], which are limited in their spatial and temporal coverage. Here we address this observational challenge and explore the impact of horizontal stirring on naturally stimulated fine-scale blooms through a detailed Lagrangian characterization of bloom development using multi-satellite products and ARGO float data. This approach allows continuous tracking of bloom dynamics and provides detailed quantification of small-scale spatio-temporal changes in bloom properties. The observations are interpreted using a simple ecosystem model, which simulates the dynamics of planktonic systems embedded within dispersing fine-scale water patches.

Potential ecological and biogeochemical consequences of tracer patch dispersion are of special interest in the oligotrophic waters of subtropical gyres, where a large part of the nutrient supply to the surface waters comes from episodic fine-scale upwelling events[12–14]. We focus our analysis on a region in the North Pacific Subtropical Gyre, where nutrient injection events at these scales have recently been documented[15,16] and where re-occurring localized summertime blooms are frequently observed in satellite data[17,18]. Satellite observations of summertime blooms in the North Pacific Subtropical Gyre, together with results from the ecosystem model, show that dispersion/dilution of fine-scale nutrient-enriched water patches may regulate the dynamics of nutrients–phytoplankton–zooplankton ecosystems and enhance accumulation of photosynthetic biomass in low-nutrient oceanic environments.

## Results

**Satellite observations.** We first focus on the evolution of one of the largest localized satellite-observed blooms, which took place in the North Pacific Subtropical Gyre during the summer of 2007 (ref. 18). The strong gradients in chlorophyll concentrations allow unambiguous delineation of bloom boundaries according to the chlorophyll contour around its periphery (Fig. 1a–d and Supplementary Fig. 1).

Detailed diagnostics of satellite images (see Supplementary Movie 1) reveals that the bloom was initiated with the emergence of elongated chlorophyll patches covering an area of $\sim 10^4\,km^2$ (Fig. 1a,b). The patches stretched, folded and merged in a continuous manner, resulting in a coherent bloom covering an area of $\sim 10^5\,km^2$ (Fig. 1d). The observed bloom occurred in late summer, when the water column is well stratified and the surface layer is depleted of nutrients[17]. This rules out the possibility that the observed patterns resulted from fine-scale stratification of the water column, which was shown to have an important role in triggering spring blooms in the North Atlantic[19]. In contrast, these summertime conditions are consistent with submesoscale upwelling of nutrient-rich waters as a trigger for bloom initiation[12,14]. This is further supported by the fact that the patches analysed here were initiated across distinct sea surface temperature fronts at the peripheries of mesoscale eddies (Supplementary Fig. 2), which are regions that favour the generation of submesocale vertical velocities[20].

The continuous change in bloom morphology suggests that the transition from fine-scale patches was driven by horizontal stirring, followed by mixing[3,5]. To investigate the effect of stirring quantitatively we conducted a numerical simulation in which synthetic particles are advected by the satellite-derived surface currents. The particles were initiated on day 232 (20 Aug 2007, Fig. 1e) according to the morphology of the bloom as delineated by the $0.13\,mg\,m^{-3}$ chlorophyll contour (Fig. 1b), and were advected by the satellite-derived surface currents until day 290 (21 October 2007; Fig. 1f), when the bloom reached its maximal areal extent (Fig. 1d). After two months of advection, the synthetic particles were dispersed by the stirring process, resulting in dilution of their density. Furthermore, spatial distribution patterns of the bio-optical (that is, chlorophyll) and numerical tracers were remarkably similar (compare Fig. 1d,f), especially considering the limited resolution of the satellite altimetry data used for advecting the synthetic particles. This similarity strongly supports the hypothesis that bloom morphology was structured primarily by horizontal stirring.

Identification of stirring as the factor structuring bloom morphology allowed the construction of unambiguous Lagrangian time series describing changes in the bloom's biophysical properties from satellite data[21] (Fig. 2). The time-varying patch surface area ($A_p$) is identified as the area bounded by a contour of equal-chlorophyll surrounding the bloom periphery (for example, the $0.13\,mg\,m^{-3}$ contour in Fig. 1a–d). As $A_p$ depends on the value of the chlorophyll contour, a sensitivity analysis of its temporal evolution was performed for a range of contour levels ($0.11$–$0.16\,mg\,m^{-3}$; Fig. 2 and Supplementary Fig. 1).

These satellite observations can be used to estimate important physical quantities related to the dispersion process, and in particular the effective eddy diffusivity ($k_e$). Throughout the patch's life time, its surface area increased linearly at an average rate of $2{,}400 \pm 370\,km^2\,d^{-1}$ ($28{,}000 \pm 4{,}300\,m^2\,s^{-1}$). The linear increase in $A_p$ is typical for dispersion occurring on a scale that is larger than the size of the local eddies[22,23]. As the plankton patch is dispersed by the turbulent oceanic flow, the rate of its expansion is a measure of $k_e$. Following Garret[22], we formulate the linear relationship between $k_e$ and patch expansion rate as follows:

$$k_e = \frac{1}{8\pi}\frac{\mathrm{d}A_p}{\mathrm{d}t} \qquad (1)$$

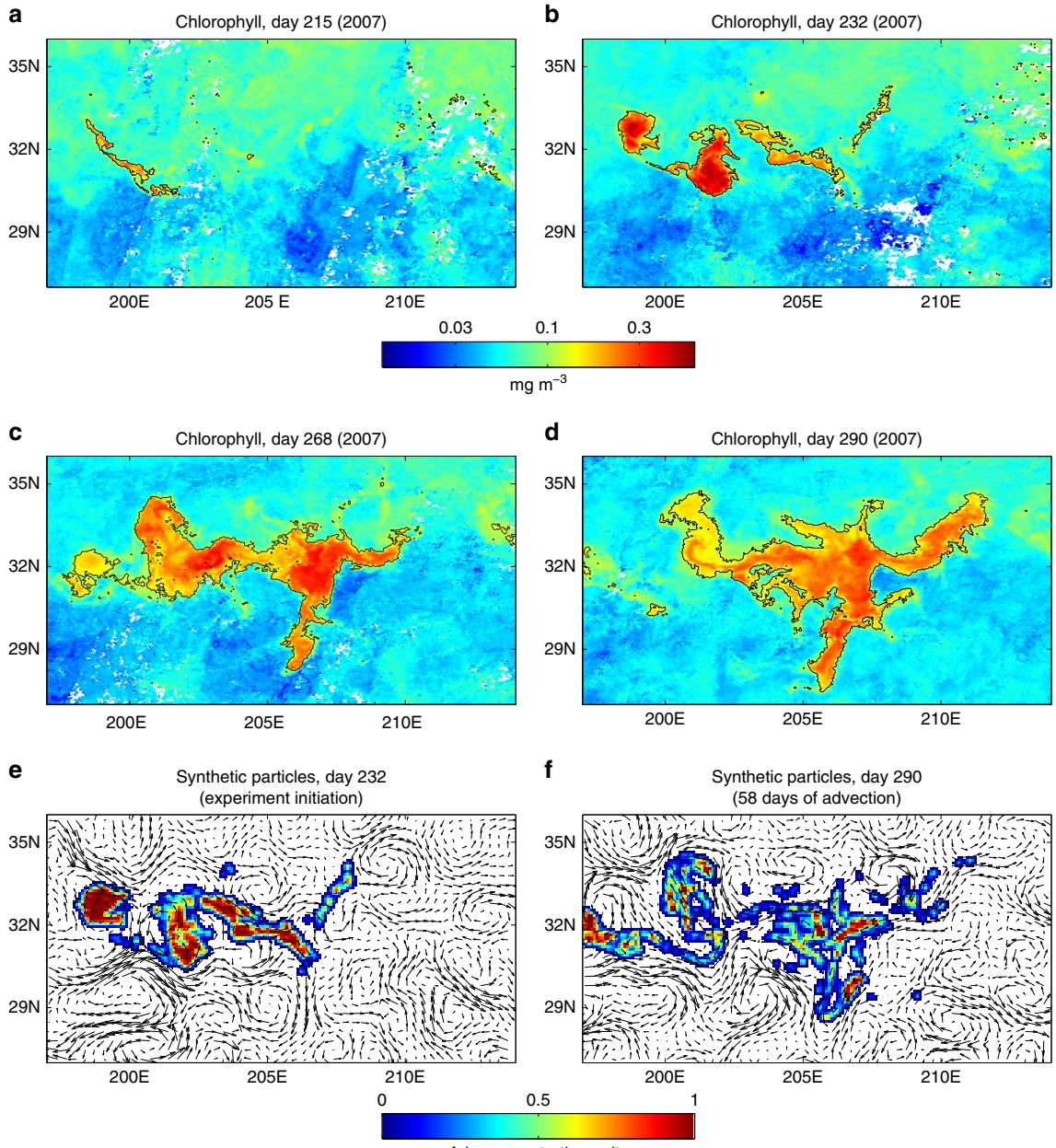

**Figure 1 | Spatio-temporal evolution of the 2007 bloom.** (**a**–**d**) Satellite-derived maps of chlorophyll concentrations showing snapshots from the 3 months bloom evolution. Black polygon delineates bloom boundaries as defined by the 0.13 mg m$^{-3}$ chlorophyll contour. (**e**,**f**) Density plot showing concentration of synthetic particles advected by the satellite-derived velocity field at the initial and final stages of the numerical experiment (days 0 and 58 of the numerical experiment, corresponding to year days 232 and 290, respectively). The particles were initiated over the area associated with the bloom's spatial extension on day 232 (that is, **b**) and were advected without any re-seeding. The positions of the particles in time is provided explicitly by the numerical integrator. Arrows represent the satellite-derived velocity vectors used for advecting the numerical particles. The remarkable similarity between bloom morphology and particle distribution after the 2-month period of particle advection is noteworthy (compare **d** and **f**).

Using this formulation, we estimate $k_e$ to be $1{,}100 \pm 150\,\mathrm{m}^2\,\mathrm{s}^{-1}$, which is consistent with previous estimates that are based on *in-situ* drifter data[24].

Satellite observations of time-varying $A_P$ were used in conjunction with mixed layer depth (MLD) from ARGO floats to quantify the mixed layer chlorophyll biomass associated with the bloom evolution. During its life-time the patch was associated with MLD of $40 \pm 10\,\mathrm{m}$, which is typical for summertime conditions in the North Pacific Subtropical Gyre[25]. Assuming the mixed layer to be homogeneous with respect to chlorophyll[26], total chlorophyll biomass was quantified

by integrating satellite-derived chlorophyll concentrations spatially over $A_P$ and vertically over the MLD. Lagrangian time series of the patch evolution shows that the increase in $A_P$ was associated with an order of magnitude increase in total chlorophyll biomass (Fig. 2b). The continuous increase in total chlorophyll biomass indicates that net growth of phytoplankton was positive during the 3-month bloom life time. In contrast to total chlorophyll biomass, averaged chlorophyll concentrations remained almost constant during most of that period (relative change of ~15% during the last 2 months; Fig. 2c).

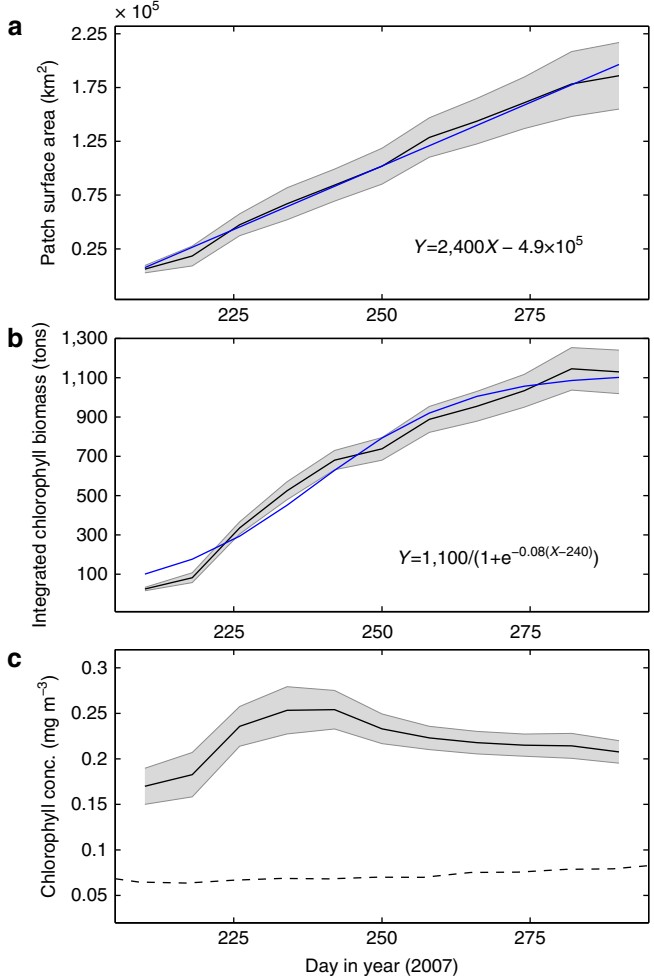

**Figure 2 | Satellite-based Lagrangian time series of the 2007 bloom.**
(**a**) Patch surface area. (**b**) Chlorophyll biomass integrated spatially over the patch surface area and vertically along the MLD as derived from ARGO floats. (**c**) Chlorophyll concentrations averaged spatially over the patch surface area. Black solid lines and shaded area denote mean and s.d. values, respectively, for different border-delimiting chlorophyll contours (0.11–0.16 mg m$^{-3}$). Blue lines in **a** and **b** represent linear and logistic fits, respectively, for the mean time series. Dashed line in **c** shows background chlorophyll concentrations.

Multi-annual analysis reveals similar patterns (that is, dispersion of fine-scale chlorophyll patches that is associated with relatively constant chlorophyll concentrations and close-to-constant increase in chlorophyll biomass accumulation) in a number of summertime blooms observed in that region (Fig. 3). Lagrangian satellite-based time series for several regional blooms indicate that the amount of particulate organic carbon (POC) accumulated in such blooms can reach the order of $10^5$ tons (Fig. 3e).

**Ecosystem model**. During its dispersion the nutrient-enriched water patch was mixed and diluted with the oligotrophic waters surrounding it. The fact that the linear increase in $A_p$ (Fig. 2a) was associated with small temporal changes in chlorophyll concentration (Fig. 2c) indicates that the losses (that is, reduction in concentration) due to dilution were compensated by biological production of phytoplankton[27]. The competition between

these two effects is especially intriguing as the biological production process is itself influenced by the dilution processes. In principle, dilution can have both a negative (due to reduction in nutrient availability) and a positive (due to reduction in concentration-dependent mortality agents such as viral infection and grazing[21,28]) impact on the phytoplankton's ability to accumulate.

We explore the effect of the dispersion/dilution process on phytoplankton biomass accumulation through an averaged-field numerical model that simulates the dynamics of a nutrient–phytoplankton–zooplankton ecosystem embedded within a water patch whose volume increases with time[29] (for a detailed model description, see the Methods section). At time $t = 0$ of the numerical experiment, the patch receives a pulse of nutrients, which represents an episodic event of fine scale nutrient injection. The nutrients are consumed by the phytoplankton, which are grazed by the zooplankton.

We assume that the observed patch is confined within the constant MLD. Accordingly, as the dispersion/dilution processes is associated with a linear increase in patch surface area, it is translated in the model to a linear increase in the volume of the patch, which is driven by a constant inflow of water that do not contain nutrients, phytoplankton and zooplankton. Such volume increase can be translated to a concentration dilution rate. Specifically, we define the horizontal dilution rate ($\beta$) as the rate of relative change in patch volume due to the dispersion/dilution processes. Based on equation (1) $\beta$ can be formulated as:

$$\beta(t) = \frac{1}{A_p(t)} 8\pi k_e \qquad (2)$$

The dilution affects all sink and source processes. For each time step, we solve for the interplay between the coupled production and removal that are regulated by the dilution, and explore the possible solution space and what can be learned from the states to which the systems converges.

We analyse the impact of horizontal dilution on the plankton ecosystem by comparing total biomass of nutrients, phytoplankton and zooplankton accumulated within the patch after the model reaches a steady state, for different initial dilution rates ($\beta_0$). A comparison between total biomass levels after the model reaches a steady state for different initial conditions reveals that the potential of a nutrient-enriched water patch to accumulate phytoplankton biomass depends on the balance between characteristic time scales of the physical and biological processes governing its dynamics. Specifically, we distinguish between three scale-dependent regimes that emerge when comparing model results for different values of the non-dimensional parameter $\gamma$, defined as the ratio between $\beta_0$ and phytoplankton growth rate ($\mu$) (Fig. 4a).

For a given grazing rate ($G$), when $\gamma \gg 1$ (that is, $\beta_0 \gg \mu$), the rapid decrease in nutrient concentration due to dilution prevents the phytoplankton from exploiting the full capacity of the nutrients reservoir and phytoplankton biomass accumulation is reduced compared with the case without dilution. In its extreme, this regime is analogous to the case of patches that are smaller than the critical size required to sustain a plankton patch, as defined by the KISS model[30,31]. When $\gamma \ll 1$ (that is, $\beta_0 \ll \mu$) phytoplankton efficiently exploit the nutrient reservoir. However, at a second stage the high phytoplankton concentration stimulates grazing, so that when the system reaches a steady state much of the phytoplankton biomass is converted into zooplankton.

In the third regime, where physical and biological time scales are comparable (that is, $\beta_0 \sim \mu$), conditions are favourable for accumulation of phytoplankton biomass. In such cases, dilution is

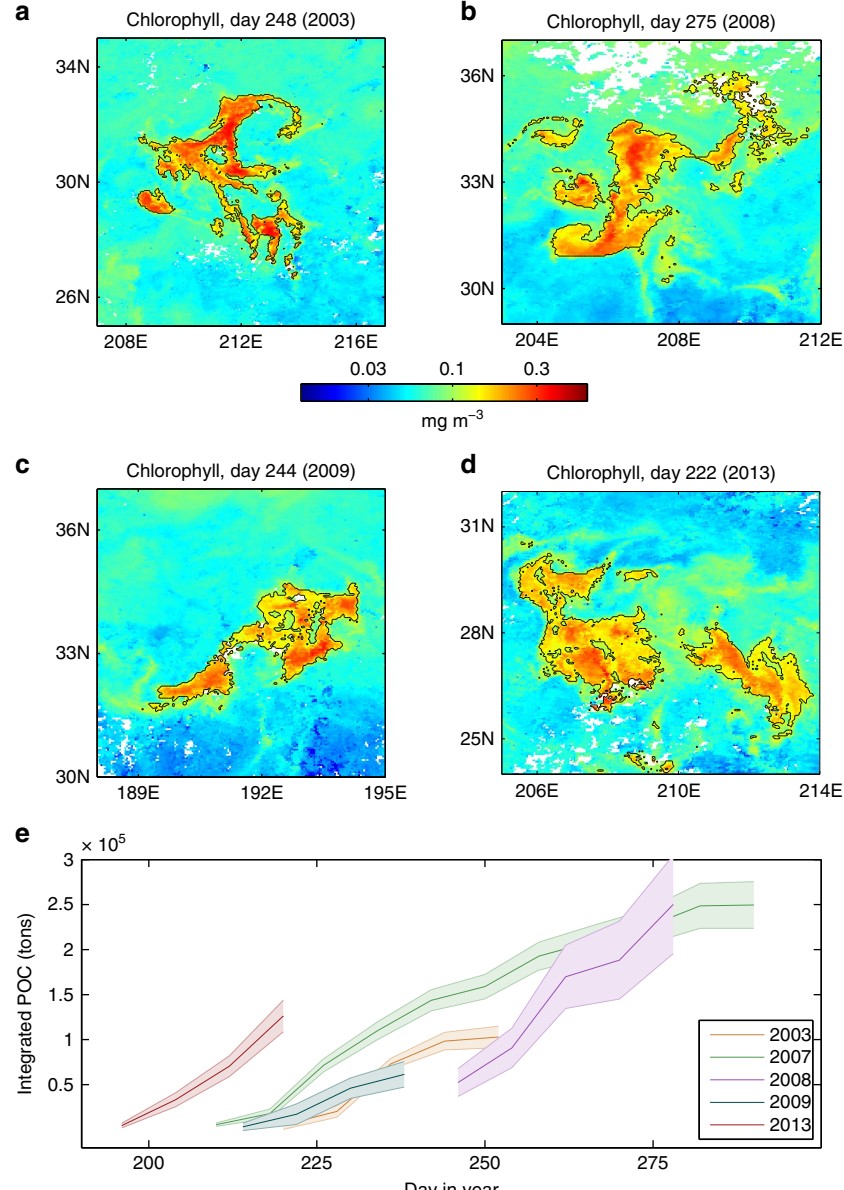

**Figure 3 | Dispersed blooms in the North Pacific Subtropical Gyre.** (a–d) Maps of satellite-derived surface chlorophyll showing snapshots from the development of fine-scale dispersed blooms observed in four different years. (e) Satellite-based Lagrangian time series of total accumulation of POC for the blooms shown in this figure and for the 2007 bloom discussed in the text.

slow enough to allow efficient utilization of the nutrient reservoir and at the same time fast enough to regulate phytoplankton concentrations such that losses due to grazing are reduced. As phytoplanktons are not starved for nutrients and are relatively protected from grazing pressure, phytoplankton biomass accumulation is enhanced. When these conditions are met, fine-scale phytoplankton patches can thrive for relatively long periods and, ultimately, form blooms that are spread over large areas.

Further examination of the interplay between physical and biological time scales is done by extracting the value of $\beta_0$ yielding maximum accumulation of phytoplankton biomass ($\beta_{0\_opt}$, e.g. black star in Fig. 4a) for model runs with different values of $\mu$ and $G$ (Fig. 4b). Comparison between the different model outputs shows that for a given grazing rate $\beta_{0\_opt}$ is linearly correlated with $\mu$. The slope of the regression line between $\beta_{0\_opt}$ and $\mu$ is proportional to the value of $G$.

## Discussion

Using satellite-based Lagrangian time series and an ecosystem model, we have shown how the interplay between governing timescales of plankton dynamics and horizontal dispersion regulates plankton blooms and enhances accumulation of photosynthetic biomass in low-nutrient environments. Owing to the stimulating effect of the dispersion/dilution process, fine-scale nutrient seeding events may be translated into relatively large ($\sim 10^5 \, km^2$) and long-lived (a few months) phytoplankton blooms.

The governing time scale of the dispersion/dilution process is the horizontal dilution rate, $\beta$, which we define as the relative change in patch surface area due to horizontal turbulent stirring and mixing. Results from a simple ecosystem model embedded within a variable-volume water patch show that ecosystem response to horizontal stirring is dictated by the balance between the physical time scale $\beta$ and biological time scales $\mu$ and

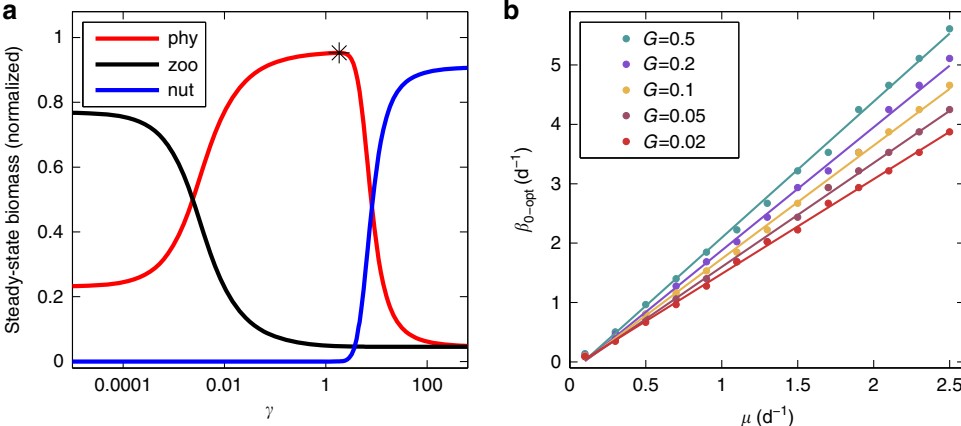

**Figure 4 | Plankton response to horizontal dilution in the ecosystem model.** (**a**) Normalized steady-state biomass accumulation of phytoplankton, zooplankton and nutrients (red, black and blue lines, respectively) for different values of the non-dimensional parameter $\gamma$, defined as the ratio between initial dilution rate and phytoplankton growth rate. Phytoplankton growth rate and grazing rate are $1.6\,\mathrm{d}^{-1}$ and $0.12\,\mathrm{d}^{-1}$, respectively. Black star marks maximum phytoplankton biomass yield (associated with $\beta_{0\_opt}$ for the specific combination of growth rate and grazing rate). (**b**) Optimal initial dilution rate as a function of phytoplankton growth rate. Colours represent different grazing rates. Solid lines show the corresponding linear fits.

$G$. Optimum conditions for phytoplankton biomass accumulation are found when $\beta_0 \sim \mu$. Under these conditions, phytoplankton can efficiently exploit the nutrient reservoir while being relatively protected from grazing pressure. Thus, by reducing losses due to concentration-dependent factors, the dispersion/dilution process can prolong the life time of fine scale blooms, which otherwise may be rapidly terminated[21].

The linkage between physical and biological time scales (Fig. 4b) also suggests that the environmental conditions associated with a given dilution rate may be preferential for specific phytoplankton species. Accordingly, fast-growing phytoplankton are expected to thrive in patches associated with high dilution rate, whereas slow-growing phytoplankton are expected to thrive in patches associated with low dilution rate. This explains, quantitatively, the partitioning of phytoplankton functional types into different turbulence regimes, as represented schematically by the mandala of Margalef[32]. Accordingly, turbulent waters that are characterized by higher values of $k_e$ (and thus, for a given patch size, higher value of $\beta$) would be preferential for faster growing phytoplankton.

Satellite observations reveal that the five dispersed North Pacific blooms discussed here (Fig. 3) are associated with $\beta_0$ values of $0.4 \pm 0.2\,\mathrm{d}^{-1}$, indicating that they are most likely to be found within the favourable regime of $\beta_0 \sim \mu$ and are thus associated with environmental conditions that allow phytoplankton to thrive.

Episodic injection events are an important source of nutrients to surface waters of oligotrophic subtropical gyres. The dispersion and consequent dilution of the resulting fine-scale phytoplankton blooms is therefore likely to be a ubiquitous phenomenon, with substantial impact on the regional ecology and biogeochemistry. As the structure of planktonic ecosystems substantially affect the flux of organic matter to the deep ocean[33], the enhancement of phytoplankton biomass accumulation by horizontal stirring may have an important impact on the efficiency of the biological carbon pump.

In contrast to the case of dilution due to vertical processes[34], the ecological and biogeochemical implications of horizontal dilution have barely been explored. Underestimation of the importance of this mechanism may result from an inherent observational limitation: because of dilution the increase in phytoplankton biomass is generally associated with only a marginal increase in surface concentration, which is the

quantity commonly derived from satellite and *in-situ* measurements. Furthermore, spatial characteristics of the dispersed fine-scale phytoplankton patches are expected to be similar to those resulting from direct cascade of blooms initiated at large scales[3–5]. Therefore, identification and quantification of the dilution process can only be achieved by tracking changes in biophysical properties of the dispersed patches in a Lagrangian manner as was done here.

Lagrangian tracking of time varying chlorophyll patches also allows retrieval of important physical quantities related to the dispersion process. Specifically, our results show good agreement between estimated values of $k_e$ (derived from the observed rate of patch expansion) and values derived from *in-situ* drifter data[24]. This indicates that dispersion patterns of chlorophyll patches can be used as a direct measure to this important physical quantity and opens the way to systematic retrieval in different parts of the World Ocean. Although there are various ways to estimate tracer dispersion and effective eddy diffusivity in the ocean[23,35–37], following chlorophyll patches in a Lagrangian manner using ocean colour satellite data provides a simple and useful observational-based platform for this matter.

Although the fact that biogeochemical properties of phytoplankton blooms may vary in response to changes in spatial distribution of nutrient-supply sources and in rates of horizontal mixing and stirring has previously been acknowledged[7–9], the critical effect on the planktonic ecosystem described here has largely been overlooked. The biological effect of horizontal dilution due to tracer patch dispersion is a sub-grid process in large-scale models that do not resolve fine-scale plankton patches. Our approach provides a recipe for parameterizing this effect, by integrating information on the typical sizes of plankton patches and on effective eddy diffusivity, which are the factors that control the horizontal dilution rate. Given the importance of horizontal dilution, such parameterization will improve our ability to represent ecological processes and biogeochemical cycles in nutrient-limited ocean basins.

## Methods

**Satellite data.** Quantification of bloom dynamics is based primarily on a detailed analysis of ocean colour satellite data. To reduce the surface area masked by clouds we integrate observations from three independent sensors: MODIS-Aqua, MODIS-Terra and MERIS. This approach allows continuous tracking of phytoplankton blooms as they are advected and deformed by the currents and provides detailed quantification

of small-scale spatio-temporal changes in bloom properties. The data set comprises 8-day composites of 4 km Level 3 data obtained from the ocean colour data distribution site (https://oceancolor.gsfc.nasa.gov/). From the ocean-colour satellite data, we extract sea surface concentrations of chlorophyll and POC. Both chlorophyll and POC are calculated using band-ratio algorithms, which are based on blue and green regions of the electromagnetic radiation spectrum. The band-ratio algorithms were tested against field measurements of chlorophyll[38] and POC[39], showing good agreement between satellite-derived and in-situ measured values.

Lagrangian time series of chlorophyll or POC are extracted by averaging the satellite-derived concentrations over the time-varying patch surface area. Total biomass accumulation is derived by integrating concentrations spatially over the patch surface area and vertically over mixed-layer depth extracted from ARGO float temperature profiles (see below). Lagrangian time series of spatially and vertically integrated POC are used as a measure to the amount of organic carbon accumulated in the course of the observed blooms.

Geostrophic surface currents were obtained from the AVISO database (http://www.aviso.oceanobs.com). For the period of our study, the distributed global product combines altimetric data from three satellite missions. The data are available on a $1/3° \times 1/3°$ Mercator grid, with one data file every 7 days. Trajectories used for Lagrangian analysis were computed by integrating the satellite-derived velocities with a Runge–Kutta scheme of the fourth order with a fixed time step of 6 h. The velocities derived from the altimetric data have been linearly interpolated in space and time. Analysis of the velocity field in a Lagrangian manner allows to simulate the formation of spatial structures that are below the resolution of the satellite altimetry data. This is made possible by the fact that small scale patterns in tracer distribution are often associated with the time varying nature of the velocity field, which is captured by Lagrangian analysis tools, Nevertheless, the limited spatial and temporal resolution of the satellite-derived velocity field, together with lack of information wind drift and ageostrophic velocities prevents perfect simulation of the stirring process, resulting in mismatches between spatial distribution patterns of modeled and observed (for example, chlorophyll) tracers.

**Argo floats.** MLD was diagnosed from Argo floats temperature profiles over a $17° \times 9°$ area (197–214°E/27–36°N) overlapping the 2007 bloom. MLD is defined as the depth at which temperature differs from a surface value by 0.2 °C. Argo floats data were collected and have been made freely available by the International Argo Program and the national programmes that contribute to it (http://www.argo.ucsd.edu and http://argo.jcommops.org). The Argo Program is part of the Global Ocean Observing System.

**Ecosystem model for a dispersed nutrient-enriched water patch.** We simulate the effect of patch dispersion and consequent dilution on plankton biomass accumulation, by implementing a simple nutrient—phytoplankton—zooplankton ecosystem model in a variable-volume reactor model[26]. At time $t = 0$, the patch receives a pulse of nutrients. Following this single event of nutrient injection, the volume of the patch increases with time in response to constant inflow of seawater that does not contain nutrients, phytoplankton and zooplankton. Assuming that the nutrient-enriched water patch is restricted to a constant depth, the inflow rate ($F$) of nutrient-free waters is equivalent to the rate of change in patch surface area, $A_p$. Following Garret[21], we formulate the time-varying $A_p$ as:

$$A_p(t) = A_p(0) + 8\pi k_e t \tag{3}$$

where $k_e$ is the effective eddy diffusivity. Accordingly, the inflow rate of nutrient-free water is parametrized as:

$$F = 8\pi k_e H_{ml} \tag{4}$$

where $H_{ml}$ is the MLD and is taken to be 40 m according to the typical summertime conditions in the North Pacific Subtropical Gyre[25].

Phytoplankton concentration is dependent on nutrient concentration and is represented by a Monod growth function. Zooplankton concentration depends on phytoplankton concentration and follows a Holling type II functional response, which assumes that the consumer is limited by its capacity to process food. At time $t = 0$, the patch contains only trace quantities of phytoplankton and zooplankton. The nutrients injected at time $t = 0$ are consumed by the phytoplankton, which are grazed by zooplankton. Concentrations of all three ecosystem components are being constantly diluted in response to the inflow of biology-free water into the patch.

The overall dynamics is described by the following equations:

$$\frac{dV}{dt} = F \tag{5}$$

$$\frac{dN}{dt} = -\mu \frac{N}{N + k_N} P - \frac{F}{V} N \tag{6}$$

$$\frac{dP}{dt} = \mu \frac{N}{N + k_N} P - \frac{F}{V} P - GZ \frac{P}{P + k_P} \tag{7}$$

$$\frac{dZ}{dt} = GZ \frac{P}{P + k_P} - \frac{F}{V} Z - mZ \tag{8}$$

where $F$ is inflow rate of nutrient-free seawater into the patch (m$^3$ d$^{-1}$), $V$ is the patch volume (m$^3$), $N$, $P$ and $Z$ are concentrations of nutrients, phytoplankton and zooplankton, respectively (mmol N m$^{-3}$), $\mu$ is the phytoplankton maximum growth rate (d$^{-1}$), $k_N$ is the nutrients half-saturation constant (mmol N m$^{-3}$), $G$ is the maximum grazing rate (d$^{-1}$) and $k_P$ is the half-saturation constant with respect to phytoplankton concentration (mmol N m$^{-3}$). Total biomass accumulation of nutrients, phytoplankton and zooplankton (Fig. 4) are derived by multiplying equations (6)–(8), respectively, by the time-varying patch volume.

The system converges into a steady-state after ∼200 days. We simulated the dynamics of phytoplankton, nutrients and zooplankton biomass, and calculated the total biomass accumulated after the system reached a steady state. The steady-state biomass of the nutrients, phytoplankton and zooplankton was normalized to the total carrying capacity of the system, which is defined as follows:

$$CC = P_0 V_0 + N_0 V_0 + Z_0 V_0 \tag{9}$$

where the subscript 0 denotes concentration and volume at time zero.

We assume that all of the biomass is conserved within the mixed layer and ignore losses due to particle sinking, transport by vertical migrators or transformation to dissolved and particulate organic matter. Phytoplankton mortality is neglected under the assumption that zooplankton do not discriminate between live and dead cells. The model does not include losses due to viral and bacterial lysis, and there is no explicit representation of the effect of temperature and light. As a first approximation, we assume uniform fields averaging of temporal and spatial variations in radiation and temperature.

The model was run using MATLAB and its built-in ode45 solver. Parameter values were similar to those used in recent ocean ecosystem model studies[40,41] (Supplementary Table 1). Sensitivity tests showed that the presented results are only weakly affected by the choice of parameters (Supplementary Fig. 3) or whether or not grazers are recycled as nutrients.

**Code availability.** The MATLAB codes used for running the ecosystem model are available for use, and can be downloaded from: https://www.dropbox.com/sh/lvdraj2oixcjr55/AAA8u6Az31jbHmfQgv28dfWca?dl=0.

**Data availability.** All the data used in this research are freely available and may be downloaded through the links detailed in the Methods section.

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

## Acknowledgements

We acknowledge support by Scott Jordan and Gina Valdez, as well as from the De Botton Center for Marine Science

## Author contributions

Y.L., I.K. and S.S. developed the Lagrangian framework for tracking the dispersed plankton patches, developed the numerical model and analysed the output. Y.L., I.K., S.S., F.D., A.V. and E.B. interpreted the data and contributed to the writing of the manuscript.

## Additional information

**Competing interests:** The authors declare no competing financial interests.

**Publisher's note**: 

