## [Peer Review File · Nature Communications]

Reviewers' comments:

Reviewer #1 (Remarks to the Author):

Review

I think this paper has some new and important ideas that are demonstrated with satellite data and a model. This review raises some concerns that I recommend be addressed. The paper needs to be improved for clarity and a number of comments and suggestions are given below to help.

Lines 48-50: Mahadevan & Campbell had showed that fine scale variance introduced by small-scale upwelling is transferred to larger scales by stirring, in contrast to the Abraham 1998 paper that treats the effects of stirring in 2D.

Line 72: "As in other cases" : Which other cases?

Line 73 : "sharp front" - do you mean the strong concentration gradient in chlorophyll? This would be less ambiguous wording, as the "front" might mean something else.

Line 74: Contour of equal chlorophyll concentration : Do you mean chlorophyll contour? Contour already conveys it is an iso-chlorophyll line.

Line 81 "the" missing

Line 82 and 85 : Reference 18 is not the right reference here.

Line 98 two month

Fig 1. d This figure could be improved. What is the lighter pink coloring? Why are the edges jagged? Are stars used for particles? How many particles? If they were seeded according to Fig 1b, could authors show (by using smaller symbols) the extent to which they concentration is diluted? Is there any re-seeding? What are the velocity fields used to stir the patch? Could the velocity field be shown? This figure leaves a lot of questions in the reader's mind. My suggest would be to replace panels a,b,c with panels similar to Fig. S1a,c and show SST in S1 instead.

Lines 93 and ensuring para: The term stirring indicates that there is no mixing, which is not the case here. In fact the horizontal diffusivity is estimated in the next section.

Fig 2. Integrating over the mixed layer depth is not justified. What is the mixed layer depth in this region? At what depth was the bloom? Nothing has been said about the vertical structure of the bloom and water column up to this point in the paper and the assumption of homogeneity of the bloom over the mixed layer, and the homogeneity of the mixed layer depth makes the estimate of "logistic" growth of biomass highly questionable.

Since the chlorophyll is growing (probably at an exponential rate) the increase in the area of the patch, which is delineated by the contour of a chlorophyll value, cannot be ascribed to just horizontal mixing. Using the area change of the chlorophyll patch (a reactive tracer) to estimate eddy diffusivity is therefore questionable. A different answer would be obtained if (in a model, for example) a conservative tracer patch was initialized and allowed to evolve. In fact the contour enclosing a high concentration would shrink as the patch evolves, and a new contour (of lessor concentration) would have to be defined to observe the increase in patch size.

There are several ways to estimate eddy diffusivity from the particles in the model. Also the tracer dispersion (with respect to its centroid) could be used. For a review of the literature on dispersion, see LaCasce, 2008. Other authors, e.g. Ledwell, Sundermeyer, Lelong have calculated dispersion from tracers.

Lien 130: What do you mean by "monotonously" ?

The growth in chlorophyll biomass cannot be explained without considering the growth and loss of

phytoplankton. But the first part of the paper draws conclusions from the increase in the size (area) of chlorophyll patches alone, and then uses those (i.e. the eddy diffusivity) in the ecosystem model in the second part.

Line 140: Constant pattern : .. but it was just shown that the patch is evolving

Page 10-11. The model needs to be better explained. I found it difficult to understand. The text describes the time-evolution of solutions for a value of "beta", while Fig 4. shows the steady state solution. Fig S2 is without much explanation, so the Suppl. Mat. did not help.

Is the dilution applied to phytoplankton, nutrient and zooplankton?

Could the results be presented in terms of some non-dim parameters, e.g. β/μ ?

Fig 4b suggests that the results could have been presented in terms of β/μ

More thought needs to be given to the figures. I think they can be more illustrative of the point that is being conveyed.

Line 200: Now the text jumps to describing an "ecological niche" . There are too many gaps in the explanation for the reader to follow.

Line 214 affects

Lines 229-231 : I have the concern described earlier - i.e. Is it justified to use a reactive tracer to estimate k_e ?

Line 227-228 - Not so remarkable - as the value of ~ 1000 m²/s seems to come up everywhere. The other independent sources should have been included in the Suppl. Mat.

Line 233 - Large scales models have a fair amount of horizontal mixing (parameterized or by other means). So I wouldn't say it is not represented. Perhaps the issue is that the smaller subgrid patches are represented in a more dilute way at the coarser scale.

Line 235 : Again, I don't think that the dilution needs to be increased in coarse models. It is the vertical injection of nutrients at small scales, and hence the initiation of the small patches that might be missing. I don't see how the addition of a dilution parameterization will help, when the patch to be diluted is itself missing from coarse models.

Methods

Much of what I complained about earlier (about not being able to understand the model) became clearer when reading this section on the modeling. Some of this needs to be clarified earlier in the paper.

Usually nitrate is expressed in m-mol/L or micro-mols per m³.

Something should be said about how/ why the very coarse resolution AVISO fields are able to capture the advection of the patch, which shows heterogeneity at much smaller scale. Some discussion can be included on what might be causing the mismatch between the particle patch and the chlorophyll observation.

SST: I couldn't see how SST was used in the work, other than in the 2 panels in Fig S1. If SST is used, why is the coarse AMSRE product chosen? Why not the 4 km GHRSSST (or even 1 km MODIS SST)?

Line 247: How is POC estimated? What algorithm? How was it used in this work?

Line 253: Three or four? The data are "available" - not gridded by the authors, I presume.

Line 254: Better to say Aviso velocities, rather than "geostrophic", because they may not be divergence free after the interpolation, I assume.

Line 256 : have been

Line 261 Argo float temperature profiles

Line 283 "As nutrients are added to the patch" - I thought that the nutrient is present at time $t=0$ to represent the impulse, and not added later.

Reviewer #2 (Remarks to the Author):

Synopsis

The authors use remote sensing data to examine a bloom (blooms) of phytoplankton in the North Pacific Subtropical Gyre. Using Lagrangian particle tracking in the geostrophic flow interpreted from altimetry, remotely sensed surface chlorophyll and information on the mixed-layer from ARGO floats they characterize the development of area, chlorophyll concentration and integrated chl-biomass. Using idealized models they interpret the development of the feature as the response to a nutrient injection event in which horizontal dilution plays a strong role in controlling the sequence. Using a simple model they explore the balance between dilution, growth and grazing rates to suggest that dilution can both retard growth (direct effect) and enhance it (by reducing the impact of grazing).

Comments

I really enjoyed this paper alot. It uses simple models and clear thinking and exploits remote and in situ data sets to develop an interesting conceptual and quantitative framework for considering the development of sporadic open ocean blooms in oligotrophic waters. It makes a great case for the value of remotes sensing products and Lagrangian diagnostics of dispersal.

I see alot of potential for this approach and I think this short manuscript lays out the concepts very clearly. I have no substantial criticisms to make - I think the paper should be published as is. It is well written, succinct and clear. I expect it to be a useful and influential addition to the literature on this topic.

Not specifically for this paper, but I wondered if the authors might also demonstrate some cases of "failed" blooms - events which never graduate to the larger scale. Could they be placed in context of the stirring/growth balance discussed here? It would be nice to see the counter case.

Reviewer #3 (Remarks to the Author):

This paper attempted to explore the role of dilution/dispersion for the generation of phytoplankton bloom in the North Pacific. Satellite-based observations on chlorophyll concentration and a simple nutrient-phytoplankton-zooplankton model were used to reproduce the bloom patterns in the North Pacific Gyres. While the topic is of interest, this reviewer is not convinced if the manuscript produced any novel results or enhanced understanding on phytoplankton bloom generation through horizontal mixing, and therefore cannot recommend publication in Nature

Communications.

Many studies showed that horizontal stirring can theoretically produce patchiness and gradients of tracers, leading to phytoplankton bloom. The authors acknowledged this in several places (e.g., abstract, page 2, and in result and discussions), but could not convince how the current study produced results different from those standard results. They mentioned about "fine-scale phytoplankton patches feeding variance back to the large-scale through dispersion", but the linear dimension of the area covered ($\sim 10^4 \text{ km}^2$) by this model was rather comparable to what they defined as "mesoscale and sub-mesoscale ($\sim 1\text{-}100 \text{ km}$, hereafter referred to as fine-scale)". Secondly, the ecosystem model they considered was rather too simple to address issues like 'phytoplankton community structure', which they mentioned in the abstract. Also, this model did not take into account the light-limited growth, and the effects of light variation were not explored. Finally, the objectives and the claims were inconsistent in several places, e.g., page 3 line 55, 'Here we address this observational challenge...', unclear how the observational challenge was addressed by a simple simulation study; they claimed in page 6 'Identification of stirring as the factor structuring bloom morphology...' or page 12 line 186 'We have shown that...horizontal stirring may regulate the community structure of planktonic ecosystems and' - how those results were new when they mentioned 'As shown in a number of model studies, in addition to structuring bloom morphology, horizontal stirring may affect different aspects of bloom ecology...' in page 2 line 39-40? Minor comments: the text could be improved in places for better clarification and consistency, e.g., page 3, line 51, 'this discrepancy is...', the discrepancy was not clearly explained; Figure 1d, 'synthetic particles...', how were those tracked? Page 14 line 217 '...implications of horizontal dilution have barely been explored..', contradicted texts in the introduction, and so on.

Below is a point-by-point response to the referees comments. Blue and black fonts correspond to the comments and to the responses, respectively.

Reviewer #1

Review

I think this paper has some new and important ideas that are demonstrated with satellite data and a model. This review raises some concerns that I recommend be addressed. The paper needs to be improved for clarity and a number of comments and suggestions are given below to help.

We thank the reviewer for the useful comments that contributed substantially to the clarity of the paper.

Lines 48-50: Mahadevan & Campbell had showed that fine scale variance introduced by small-scale upwelling is transferred to larger scales by stirring, in contrast to the Abraham 1998 paper that treats the effects of stirring in 2D.

We thank the reviewer for this point. Following this comment we re-wrote the paragraph as follows (lines 46-49):

“The opposite scenario of fine-scale phytoplankton patches feeding variance back to larger scales through dispersion is also expected¹¹. However, the process and its impact on the planktonic ecosystem have largely been overlooked.”

Line 72: “As in other cases” : Which other cases?

The phrase is redundant and was omitted from the manuscript.

Line 73 : “sharp front” - do you mean the strong concentration gradient in chlorophyll? This would be less ambiguous wording, as the “front’ might mean something else.

We agree with the reviewer and replaced “*sharp front*” with “*strong gradient in chlorophyll concentrations*”.

Line 74: Contour of equal chlorophyll concentration : Do you mean chlorophyll contour? Contour already conveys it is an iso-chlorophyll line.

We replaced “*Contour of equal chlorophyll concentration*” with “*chlorophyll contour*”.

Line 81 “the” missing

Following the comment below, the sentence was changed as follows (line 80-83):“*This rules out the possibility that the observed patterns resulted from column's stratification¹⁸ ter wa fine scale stratification of the water column, which was shown to have an important role in triggering spring blooms in the North Atlantic¹⁹”*

Line 82 and 85 : Reference 18 is not the right reference here.

The reference in line 82 refers to the processes of bloom initiation through meososcale water column

stratification that was recently reported by Mahadevan et al. 2012 (reference 19 in the revised manuscript). This is now explicitly explained in the text (lines 80-83):

“This rules out the possibility that the observed patterns resulted from fine scale stratification of the water column, which was shown to have an important role in triggering spring blooms in the North Atlantic¹⁹”.

The reference in line 85 was indeed incorrect, and was replaced by reference 20 (Calil et al., 2012).

Line 98 two month

The typo was corrected.

Fig 1. This figure could be improved. What is the lighter pink coloring? Why are the edges jagged? Are stars used for particles? How many particles? If they were seeded according to Fig 1b, could authors show (by using smaller symbols) the extent to which they concentration is diluted? Is there any re-seeding? What are the velocity fields used to stir the patch? Could the velocity field be shown? This figure leaves a lot of questions in the reader’s mind. My suggest would be to replace panels a,b,c with panels similar to Fig. S1a,c and show SST in S1 instead.

We thank the reviewer for these helpful comments. Following these comments we modified the figure and its caption. As shown below, the figure is now composed of 6 panels, which can be divided into two parts: 1. panels a-d show stages in the bloom evolution as observed in the satellite-derived chlorophyll data. 2. panels e-f show the particle distribution at the initial and final stages of the numerical experiment, respectively. Superimposed on panels e-f are arrows showing the satellite-derived velocity vectors used for advecting the particle in experiment (we find that superimposing the velocity vectors in the chlorophyll images reduces substantially the clarity of the figure).

We agree with the reviewer that presenting the results from the numerical experiment in a manner the captures the effect of dilution is very useful. To show it we use a density plot, which gives a measure to the spatial concentration of particle at each grid point. In addition, the following sentence was added to the manuscript (lines 110-112):

“After two months of advection the synthetic particles were dispersed by the stirring process, resulting dilution of their density. ”.

According to these modifications, and to improve the clarity of the figure, the figure caption was re-written as well (see below).

“Figure 1. Spatio-temporal evolution of the 2007 bloom. (a-d) Satellite-derived maps of chlorophyll concentrations showing snapshots from the 3 months bloom evolution. Black polygon delineates bloom boundaries as defined by the 0.13 mg m^{-3} chlorophyll contour. (e-f) Density plot showing concentration of synthetic particles advected by the satellite-derived velocity field at the initial and final stages of the numerical experiment (days 0 and 58 of the numerical experiment, corresponding to year days 232 and 290, respectively). The particles were initiated over the area associated with the bloom’s spatial extension on day 232 (i.e. Fig. 1b), and were advected without any re-seeding. The positions of the particles in time is provided explicitly by the numerical integrator. Arrows represent the satellite-derived velocity vectors used for advecting the numerical particles. Note the remarkable similarity between bloom morphology and particle distribution after the two-month period of particle advection (compare panels d and f).”

Lines 93 and ensuring para: The term stirring indicates that there is no mixing, which is not the case here. In fact the horizontal diffusivity is estimated in the next section.

We agree with the reviewer that the point of stirring and mixing should be discussed in a more precise manner. The morphology of the patch is controlled by the combined effect of stirring and mixing: while stirring acts to disperse the patch over a large area while stretching and filamenting it, mixing acts to reduce the gradients and to homogenize phytoplankton distribution. To improve the accuracy and clarity of the manuscript, we have made the following modifications: In the Introduction section (line 38) we replaced the discussion on “*stirring processes*” with a discussion on the “*horizontal redistribution process*”, which includes the combined effect of stirring and mixing. We then point to the specific effects of stirring (stretching and filamenting a tracer patch) and mixing (homogenizing the distribution). In addition, throughout the Results section we discuss explicitly the combined effect of stirring and mixing. For example (lines 104-105):

“The continuous change in bloom morphology suggests that the transition from fine-scale patches was driven by horizontal stirring, followed by mixing^{3,5}.”

Fig 2. Integrating over the mixed layer depth is not justified. What is the mixed layer depth in this region? At what depth was the bloom? Nothing has been said about the vertical structure of the bloom and water column up to this point in the paper and the assumption of homogeneity of the bloom over the mixed layer, and the homogeneity of the mixed layer depth makes the estimate of “logistic” growth of biomass highly questionable.

We thank the reviewer for drawing our attention to this important point. As mentioned in the Methods section, information on the mixed layer depth is derived from ARGO float temperature profiles at the vicinity of the bloom. This is now mentioned explicitly in the Results section, together with a description of the mixed layer depths that were observed during the bloom and their association with climatological depths in the region. Our Lagrangian analysis of total chlorophyll biomass is based on the assumption that the mixed layer is homogeneous with respect to chlorophyll (see for example Zawaba et al. ,2005 that was added to the manuscript). Specifically, to address these issues we re-wrote the relevant paragraph in the Results section as follows (lines 134-139):

“Satellite observations of time-varying A_p were used in conjunction with mixed layer depth (MLD) from ARGO floats to quantify the mixed layer chlorophyll biomass associated with the bloom evolution. During its life-time, the patch was associated with MLD of 40 ± 10 m, which is typical for summertime conditions in the North Pacific Subtropical Gyre²⁵. Assuming the mixed layer to be homogeneous with respect to chlorophyll²⁶, total chlorophyll biomass was quantified by integrating satellite-derived chlorophyll concentrations spatially over A_p , and vertically over the MLD.”

Since the chlorophyll is growing (probably at an exponential rate) the increase in the area of the patch, which is delineated by the contour of a chlorophyll value, cannot be ascribed to just horizontal mixing. Using the area change of the chlorophyll patch (a reactive tracer) to estimate eddy diffusivity is therefore questionable. A different answer would be obtained if (in a model, for example) a conservative tracer patch was initialized and allowed to evolve. In fact the contour enclosing a high concentration would shrink as the patch evolves, and a new contour (of lessor concentration) would have to be defined to observe the increase in patch size.

There are several ways to estimate eddy diffusivity from the particles in the model. Also the tracer

dispersion (with respect to its centroid) could be used. For a review of the literature on dispersion, see LaCasce, 2008. Other authors, e.g. Ledwell, Sundermeyer, Lelong have calculated dispersion from tracers.

We thank the reviewer for noting these important references that were added to the manuscript. We agree with the reviewer's remark about the caveats of using contours of a reactive tracer for estimating eddy diffusivity. However, in the case reported here, during the patch lifetime the chlorophyll concentrations decrease very sharply around its periphery, providing a clear marker to patch boundaries and allowing unambiguous delineation of the area occupied by the tracer. To make this point more evident, we have added chlorophyll cross sections to the Supplementary Materials (Supplementary Fig. 1, see below), showing that patch surface area depends only marginally on the threshold used for delineating its boundaries. This is explained by the fact chlorophyll concentrations change only little during the patch's life time, due to the existence of a balance between net biological growth and horizontal dilution (see text), keeping a distinct border between the patch and the poor Chl background.

We acknowledge the fact that there are various ways to estimate effective eddy diffusivity in the ocean. Nevertheless, we find that despite the inherent limitations of interpreting the dispersion a reactive tracer, using chlorophyll for this matter has substantial advantages, as it is a ubiquitous tracer that is easily detected by ocean-color satellites. Although chlorophyll is not a passive tracer, its spatial distribution is strongly affected by stirring and mixing processes, indicating that it can be used as a useful marker for quantifying effective eddy diffusivity in different parts of the ocean. To emphasize this point we have added the following sentence (lines 307-309):

“Although there are various ways to estimate tracer dispersion and effective eddy diffusivity in the ocean^{23,35-37}, following chlorophyll patches in a Lagrangian manner using ocean color satellite data provides a simple and useful observational-based platform for this matter.”.

Supplementary Figure 1. Spatio-temporal evolution and patch boundaries of the 2007 bloom. *Left panels:* Satellite-derived maps of chlorophyll concentrations showing snapshots from the 3 months bloom evolution. Black polygon delineates bloom boundaries as defined by the 0.13 mg m^{-3} chlorophyll contour. White box delineates the area covered by the cross-section on the right. *Right panels:* Cross-patch sections of satellite-derived chlorophyll. Red lines mark lowest and highest chlorophyll levels used for delineating patch boundaries (Fig. 2). The figure emphasizes the strong gradients surrounding the patch, which allow unambiguous delineation of its dispersion.

Line 130: What do you mean by “monotonously” ?

We replaced “*monotonously*” with “*close to constant*”.

The growth in chlorophyll biomass cannot be explained without considering the growth and loss of phytoplankton. But the first part of the paper draws conclusions from the increase in the size (area) of chlorophyll patches alone, and then uses those (i.e. the eddy diffusivity) in the ecosystem model in the second part.

This is an important point, and following this comment we did our best to make it clearer in the revised manuscript. The increase in chlorophyll biomass results, directly, from net growth of phytoplankton.

To clarify this point, the following sentence was added (lines 141-143):

“The continuous increase in total chlorophyll biomass indicates that net growth of phytoplankton was positive during the three month bloom life time .”

From observational view point, we identify the increase in biomass from the fact that the phytoplankton patch increased in size while maintaining close-to-constant chlorophyll concentrations (Fig. 2). Then, using the ecosystem model, we show that the increase in patch size, while maintaining the same concentration, stimulates the accumulation of phytoplankton biomass due to the positive affect of the horizontal dilution (Fig. 4). The linkage between phytoplankton growth and increase in patch surface area is described as follows (lines 180-183):

“During its dispersion the nutrient-enriched water patch was mixed and diluted with the oligotrophic waters surrounding it. The fact that the linear increase in A_p (Fig. 2c) was associated with small temporal changes in chlorophyll concentration (Fig. 2c) indicates that the losses (i.e. reduction in concentration) due to dilution were compensated by biological production of phytoplankton²⁷.”

Line 140: Constant pattern : .. but it was just shown that the patch is evolving

The term “constant pattern” referred to the evolution in time of chlorophyll concentrations averaged spatially over the patch surface area, which shows little variability (unlike patch morphology and total chlorophyll biomass). To clarify this point, we replaced the phrase “*the nearly constant pattern in chlorophyll concentration*” with “*the small temporal change in chlorophyll concentration (Fig. 2c)*”.

Page 10-11. The model needs to be better explained. I found it difficult to understand. The text describes the time-evolution of solutions for a value of “beta”, while Fig 4. shows the steady state solution. Fig S2 is without much explanation, so the Suppl. Mat. did not help. Is the dilution applied to phytoplankton, nutrient and zooplankton? Could the results be presented in terms of some non-dim parameters, e.g. β/μ ?

Following the reviewer's comment we have made efforts to improve the clarity of the model description. Importantly we emphasize that the model results are discussed in terms of total biomass levels reached after the bloom converted to a steady state, and that the dilution is applied to phytoplankton, zooplankton and nutrients. To improve its clarity, the description of the model in the Results section was re-written as follows (lines 188-207):

“We explore the effect of the dispersion/dilution process on phytoplankton biomass accumulation

through an averaged-filed numerical model that simulates the dynamics of a nutrient – phytoplankton – zooplankton ecosystem embedded within a water patch whose volume increases with time²⁹ (for a detailed model description see the Methods section). At time $t=0$ of the numerical experiment the patch receives a pulse of nutrients, which represents an episodic event of fine scale nutrient injection. The nutrients are consumed by the phytoplankton, which are grazed by the zooplankton.

We assume that the observed patch is confined within the constant MLD. Accordingly, as the dispersion/dilution processes is associated with a linear increase in patch surface area, it is translated in the model to a linear increase in the volume of the patch, which is driven by a constant inflow of water that do not contain nutrients, phytoplankton and zooplankton. Such volume increase can be translated to a concentration dilution rate. Specifically, we define the horizontal dilution rate (β) as the rate of relative change in patch volume due to the dispersion/dilution processes. Based on Equation (1) β can be formulated as:

$$\beta(t) = \frac{1}{A_p(t)} 8 \pi k_e \quad (2)$$

The dilution affects all sink and source processes. For each time step we solve for the interplay between the coupled production and removal that are regulated by the dilution, and explore the possible solution space and what can be learned from the states to which the systems converges.

We analyze the impact of horizontal dilution on the plankton ecosystem by comparing total biomass of nutrients, phytoplankton and zooplankton accumulated within the patch after the model reaches a steady state, for different initial dilution rates (β_0)”

In addition, following the reviewer's suggestion, the model results are now discussed in terms of the non-dimensional parameter γ that is defined as the ratio between initial horizontal dilution rate and phytoplankton growth rate (lines 211-213):

“Specifically, we distinguish between three scale-dependent regimes that emerge when comparing model results for different values of the non-dimensional parameter γ , defined as the ratio between β_0 and phytoplankton growth rate (μ) (Fig. 4a)”.

Figure 4. Plankton response to horizontal dilution in the ecosystem model. (a) Normalized steady-state biomass accumulation of phytoplankton, zooplankton and nutrients (red, black and blue lines, respectively) for different values of the non-dimensional parameter γ , defined as the ratio between initial dilution rate and phytoplankton growth rate. Phytoplankton growth rate and grazing rate are $1.6 d^{-1}$ and $0.12 d^{-1}$, respectively. Black star marks maximum phytoplankton biomass yield (associated with $\beta_{0,opt}$ for the specific combination of growth rate and grazing rate). **(b)** Optimal initial dilution rate as a function of phytoplankton growth rate. Colors represent different grazing rates. Solid lines show the corresponding linear fits.

Fig 4b suggests that the results could have been presented in terms of β/μ

As noted above, following the reviewer's suggestion we present and discuss the model results in terms of the nondimensional parameter γ , defined as the ratio β/μ .

More thought needs to be given to the figures. I think they can be more illustrative of the point that is being conveyed.

We have made efforts to improve the figures and their description (in the main text and in the figure captions). Importantly, following the reviewer's suggestion we now show the results using the nondimensional parameter γ that is defined as the ratio between initial horizontal dilution rate and phytoplankton growth rate. In addition, the description of Fig. 4b was re-written as follows (lines 230-235):

“Further examination of the interplay between physical and biological time scales is done by extracting the value of β_0 yielding maximum accumulation of phytoplankton biomass (β_{0_opt} , e.g. black star in Fig. 4a) for model runs with different values of μ and G (Fig. 4b). Comparison between the different model outputs shows that for a given grazing rate β_{0_opt} is linearly correlated with μ . The slope of the regression line between β_{0_opt} and μ is proportional to the value of G .”

Line 200: Now the text jumps to describing an “ecological niche”. There are too many gaps in the explanation for the reader to follow.

Following the reviewer comment we deleted this term from the text. The point we intended to convey is that our results suggest that the environmental conditions associated with a given dilution rate may be preferential for specific phytoplankton. Specifically, fast-growing phytoplankton are expected to thrive in patches associated with high dilution rate, whereas slow-growing phytoplankton are expected to thrive in patches associated with low dilution rate. In order to improve the clarity and accuracy of the text, the paragraph was modified as follows (lines 265-269):

“The linkage between physical and biological time scales (Fig. 4b) also suggests that the environmental conditions associated with a given dilution rate may be preferential for specific phytoplankton species. Accordingly, fast-growing phytoplankton are expected to thrive in patches associated with high dilution rate, whereas slow-growing phytoplankton are expected to thrive in patches associated with low dilution rate.”

Line 214 affects

The typo was corrected.

Lines 229-231 : I have the concern described earlier - i.e. Is it justified to use a reactive tracer to estimate k_e ?

Although chlorophyll is not a passive tracer, its spatial distribution are often driven by mixing processes. Therefore, Lagrangian tracking of the development of chlorophyll patches can be used for estimating effective diffusivity. Specifically, this approach is likely to be useful in cases where chlorophyll concentrations within the patch are close to constant, and especially in regions where fine scale nutrient-enriched water patches are embedded within low-nutrient environments. In such cases one can expect a distinct boundary between the patch and its surrounding, and thus an unambiguous

delineation of the area occupied by the dispersed tracer.

Line 227-228 - Not so remarkable - as the value of ~ 1000 m²/s seems to come up everywhere. The other independent sources should have been included in the Supp. Mat.

We replaced “*remarkable agreement*” with “*good agreement*” and “*Other, independent sources*” with “*in-situ drifter data*²³”.

Line 233 - Large scales models have a fair amount of horizontal mixing (parameterized or by other means). So I wouldn't say it is not represented. Perhaps the issue is that the smaller subgrid patches are represented in a more dilute way at the coarser scale.

We agree with the reviewer that horizontal mixing itself is well represented in large scale models. However, the impact of horizontal mixing on fine scale patches is often a sub-grid process that is not well represented. Our results suggest that this sub-grid process may have an important, yet under-represented, impact on the ecosystem. Accordingly, for a given amount of nutrients, the structure of the plankton ecosystem within a model grid-point may vary in response to changes in phytoplankton spatial distribution patterns and horizontal mixing (i.e. the factors that control the horizontal dilution rate). This effect can be parameterized, for example, by ascribing information on the typical sizes of phytoplankton patches and on the effective eddy diffusivity. To improve the clarity of this point, the paragraph was re-written as follows (lines 313-317):

“The biological effect of horizontal dilution due to tracer patch dispersion is a sub-grid process in large-scale models that do not resolve fine-scale plankton patches. Our approach provides a recipe for parameterizing this effect, by integrating information on the typical sizes of plankton patches and on effective eddy diffusivity, which are the factors that control the horizontal dilution rate”.

Line 235 : Again, I don't think that the dilution needs to be increased in coarse models. It is the vertical injection of nutrients at small scales, and hence the initiation of the small patches that might be missing. I don't see how the addition of a dilution parameterization will help, when the patch to be diluted is itself missing from coarse models.

As we now mention explicitly in the text, in large scale models that do not resolve fine scale plankton patches the effect of horizontal dilution on the planktonic ecosystem is a sub-grid process. Since this process can alter plankton food-web structure, neglecting it may reduce the accuracy of plankton representation in such models. For example, for a given amount of nutrients within a given grid point of the model, biomass of phytoplankton and zooplankton may differ in response to changes in levels of plankton patchiness. This effect can be parametrized by including information on typical sizes plankton patches and on effective eddy diffusivity at each grid point.

Methods

Much of what I complained about earlier (about not being able to understand the model) became clearer when reading this section on the modeling. Some of this needs to be clarified earlier in the paper.

We have elaborated the description of the model in the Results part. For a detailed description of the changes please see our reply above and the revised text in lines (188-207).

Usually nitrate is expressed in m-mol/L or micro-mols per m³.

We have changed the units accordingly.

Something should be said about how/ why the very coarse resolution AVISO fields are able to capture the advection of the patch, which shows heterogeneity at much smaller scale. Some discussion can be included on what might be causing the mismatch between the particle patch and the chlorophyll observation.

We added the following paragraph to the Method section (lines 346-352):

“Analysis of the velocity field in a Lagrangian manner allows to simulate the formation of spatial structures that are below the resolution of the satellite altimetry data. This is made possible by the fact that small scale patterns in tracer distribution are often associated with the time varying nature of the velocity field, which is captured by Lagrangian analysis tools, Nevertheless, the limited spatial and temporal resolution of the satellite-derived velocity field, together with lack of information wind drift and ageostrophic velocities prevents perfect simulation of the stirring process, resulting in mismatches between spatial distribution patterns of modeled and observed (e.g. chlorophyll) tracers”.

SST: I couldn't see how SST was used in the work, other than in the 2 panels in Fig S1. If SST is used, why is the coarse AMSRE product chosen? Why not the 4 km GHRSSST (or even 1 km MODIS SST)? Indeed SST is only used in the Supplementary Figure. Accordingly, we moved the description of the SST data to the caption of this figure. Following the reviewer's comments, we now make use of 4km SST data from MODIS-Aqua.

Line 247: How is POC estimated? What algorithm? How was it used in this work?

We have added the following paragraphs to the the Methods section (lines 329-333 and 337-339):

“Both chlorophyll and POC are calculated using band-ratio algorithms, which are based on blue and green regions of the electromagnetic radiation spectrum. The band-ratio algorithms were tested against field measurements of chlorophyll³⁸ and POC³⁹, showing good agreement between satellite-derived and in-situ measured values.”.

and

“Lagrangian time series of spatially and vertically integrated POC are used as a measure to the amount of organic carbon accumulated in the course of the observed blooms.”.

Line 253: Three or four? The data are “available” - not gridded by the authors, I presume.

We corrected the manuscript, specifying that the product data is comprised of three satellites, and that it is “available”.

Line 254: Better to say Aviso velocities, rather than “geostrophic”, because they may not be divergence free after the interpolation, I assume.

We replaced “*geostrophic*” with “*satellite-derived*”.

Line 256 : have been

We added “*have been*” to the text.

Line 261 Argo float temperature profiles

The typo was corrected.

Line 283 “As nutrients are added to the patch” - I thought that the nutrient is present at time $t=0$ to represent the impulse, and not added later.

Indeed nutrients are introduced only once, at the beginning of the model run. To clarify this point, we replaced “*As the nutrients are added to the patch, they...*” with “*The nutrients injected at time $t = 0$ are...*”.

Reviewer #2

Synopsis

The authors use remote sensing data to examine a bloom (blooms) of phytoplankton in the North Pacific Subtropical Gyre. Using Lagrangian particle tracking in the geostrophic flow interpreted from altimetry, remotely sensed surface chlorophyll and information on the mixed-layer from ARGO floats they characterize the development of area, chlorophyll concentration and integrated chl-biomass. Using idealized models they interpret the development of the feature as the response to a nutrient injection event in which horizontal dilution plays a strong role in controlling the sequence. Using a simple model they explore the balance between dilution, growth and grazing rates to suggest that dilution can both retard growth (direct effect) and enhance it (by reducing the impact of grazing).

Comments

I really enjoyed this paper a lot. It uses simple models and clear thinking and exploits remote and in situ data sets to develop an interesting conceptual and quantitative framework for considering the development of sporadic open ocean blooms in oligotrophic waters. It makes a great case for the value of remote sensing products and Lagrangian diagnostics of dispersal.

I see a lot of potential for this approach and I think this short manuscript lays out the concepts very clearly. I have no substantial criticisms to make - I think the paper should be published as is. It is well written, succinct and clear. I expect it to be a useful and influential addition to the literature on this topic.

Not specifically for this paper, but I wondered if the authors might also demonstrate some cases of "failed" blooms - events which never graduate to the larger scale. Could they be placed in context of the stirring/growth balance discussed here? It would be nice to see the counter case.

We thank the reviewer for the positive and encouraging feedback. As suggested by the reviewer, in a previous work we have observed a plankton bloom that was confined within the core of a mesoscale eddy and did not disperse. The dynamics of that bloom was characterized by a rapid demise that, based on in-situ measurements, was attributed to viral infection. Following the reviewer's comment, we added the following sentence to the manuscript (lines 262-264):

“Thus, by reducing losses due to concentration-dependent factors, the dispersion/dilution process can prolong the life time of fine scale blooms, which otherwise may be rapidly terminated²¹.”

References

Lehahn, Y., et al. Decoupling physical from biological processes to assess the impact of viruses on a mesoscale algal bloom. *Curr. Biol.* **24**, doi: 10.1016/j.cub.2014.07.046 (2014).

Reviewer #3

This paper attempted to explore the role of dilution/dispersion for the generation of phytoplankton bloom in the North Pacific. Satellite-based observations on chlorophyll concentration and a simple nutrient-phytoplankton-zooplankton model were used to reproduce the bloom patterns in the North Pacific Gyres. While the topic is of interest, this reviewer is not convinced if the manuscript produced any novel results or enhanced understanding on phytoplankton bloom generation through horizontal mixing, and therefore cannot recommend publication in Nature Communications. Many studies showed that horizontal stirring can theoretically produce patchiness and gradients of tracers, leading to phytoplankton bloom. The authors acknowledged this in several places (e.g., abstract, page 2, and in result and discussions), but could not convince how the current study produced results different from those standard results.

We thank the reviewer for the efforts and comments. Indeed, as acknowledged in the manuscript, the effect of horizontal stirring on phytoplankton blooms was subject to a large number of studies. However, the work presented here differs substantially from the main body of work on this subject, and provide a novel perspective on the interaction between plankton ecosystem and the oceanic flow in which they are embedded. As noted by the reviewer, studies of the effect of stirring on phytoplankton bloom focus on the production of patchiness and enhancement of tracer gradients. This is the well known direct cascade process (Abraham, 1998) that was subject to numerous observational and theoretical studies. Unlike the previous studies, here we describe the biological consequences of the opposite process, namely the homogenization of tracer distribution and the reduction of phytoplankton patchiness by the stirring and mixing process. We show that this processes has a critical effect on plankton systems.

A highly important consequence of the observed dispersion/dilution process is the substantial enhancement of phytoplankton blooms in low-nutrients environments. We show that phytoplankton blooms initiated by fine-scale nutrient-injection events can sustain production of organic matter over periods of several months. Moreover, we show that the ratio between the biological and physical characteristic time scales describes well the solution for which the system will converge to. When the biological time scale of phytoplankton growth rate and the physical time scale of the newly-defined horizontal dilution rate are of the same order, conditions are favorable for accumulation of photosynthetic biomass. Whereas when the dilution rate is much higher or much lower than phytoplankton growth rate, photosynthetic biomass accumulation will be lower, due to inability to efficiently exploit the nutrient reservoir or due to stimulated grazing pressure, respectively

The insights achieved in this work were made possible by applying Lagrangian analysis methods and extraction of Lagrangian time series from satellite data, which allow tracking changes in biophysical properties of plankton patches as they are advected and dispersed by the ocean currents. Systematic application of such analysis methods, which are currently not commonly used, open new avenues for satellite data interpretation and altogether for better understanding of the biophysical interactions in the ocean.

They mentioned about "fine-scale phytoplankton patches feeding variance back to the large-scale through dispersion", but the linear dimension of the area covered ($\sim 10^4$ km²) by this model was rather comparable to what they defined as "mesoscale and sub-mesoscale (~ 1 -100 km, hereafter referred to as fine-scale)".

We agree with the reviewer that the use of the absolute term "*large scale*" when discussing the effect of dispersion is not precise, and we replace it with the relative term "*larger scales*" that refers to any substantial increase in patch area. The area covered by the plankton patch described here increased due to dispersion by almost an order of magnitude (from $\sim 10^4$ km² in its initial phase to $\sim 10^5$ km² in its final phase, Fig. 2a), which is in accordance with this notion.

Secondly, the ecosystem model they considered was rather too simple to address issues like 'phytoplankton community structure', which they mentioned in the abstract. Also, this model did not take into account the light-limited growth, and the effects of light variation were not explored.

When using the term "*plankton community structure*" we referred to the relative abundance of autotrophic and heterotrophic components of the planktonic ecosystem, which indeed changes in response to changes in horizontal dilution rate in the model used here. We acknowledge that the use of this term may be miss-interpreted, and replaced it with the more accurate term "*plankton food-web structure*"

Our objectives were focused on describing the effect of dilution on the food-web structure of planktonic ecosystem. To do so we developed an ecosystem model for which the maximum phytoplankton growth rate is parametrized by a constant, which can also be represented as a function of light and temperature (Dutkiewicz et al., 2009). Since the model is meant to simulate blooms associated with constrained summertime conditions in subtropical gyres, as a first approximation we averaged temporal and spatial variations in radiation and temperature, assuming that any modification of the maximum growth rate due to light-limiting and temperature-limiting conditions will be homogeneous throughout the patch. Moreover in this approximation self-shading is also averaged in. To clarify this point, we added the following text to the Methods section (lines 409-410):

"The effect of temperature and light is parametrized by the constant μ . As a first approximation we assume uniform fields averaging of temporal and spatial variations in radiation and temperature".

Finally, the objectives and the claims were inconsistent in several places, e.g., page 3 line 55, 'Here we address this observational challenge...', unclear how the observational challenge was addressed by a simple simulation study; they claimed in page 6 'Identification of stirring as the factor structuring bloom morphology...' or page 12 line 186 'We have shown that...horizontal stirring may regulate the community structure of planktonic ecosystems and' - how those results were new when they mentioned 'As shown in a number of model studies, in addition to structuring bloom morphology, horizontal stirring may affect different aspects of bloom ecology...' in page 2 line 39-40?

We thank the reviewer for highlighting places in which our description should be clearer. We will address the above comments point by point. We note that the results reported in this manuscript consists of two parts: 1. Detailed analysis of observations from satellite remote sensing data and ARGO float *in-situ* data and 2. A first approximation numerical model of a plankton ecosystem that captures the interplay between, horizontal dilution and ecological dynamics.

The satellite observations of fine scale bloom dispersion, and the satellite-based quantification of time-varying biophysical properties associated with it are, to our knowledge, quite unique. These observations were made possible by application of Lagrangian analysis methods that have only recently been developed, which allows addressing the challenge of tracking fine scale chlorophyll patches as they are advected and dispersed by the ocean currents. To clarify this point, the use of satellite and in-situ observations is now more clearly introduced (lines 55-58):

“Here we address this observational challenge, and explore the impact of horizontal stirring on naturally-stimulated fine-scale blooms through a detailed Lagrangian characterization of bloom development using multi-satellite products and ARGO float data.”.

Indeed, as we acknowledge in the manuscript, many aspects of the effect of horizontal stirring on phytoplankton have been previously reported. Specifically, the identification of stirring as the primary factor responsible for structuring bloom morphology is not new and citations are provided to acknowledge previous studies on this topic (lines 104-105):

“The continuous change in bloom morphology suggests that the transition from fine-scale patches is was driven by horizontal stirring, followed by mixing^{3,5}”

The observational and model-based understanding showing that by diluting fine scale nutrient-enriched patches with low-nutrient ambient waters horizontal stirring can enhance accumulation of phytoplankton biomass is to the best of our knowledge new. As noted above, we acknowledge that using the term “*plankton community structure*” for discussing the described effect may lead to misinterpretation of the text, and it was omitted. We agree that in its original version, the first paragraph of the Discussion section (page 12 line 186) was written in a way that can lead to misinterpretation of key messages of the paper, and it was re-written as follows (lines 245-249):

“Using satellite-based Lagrangian time series and a simple ecosystem model we have shown how the interplay between governing timescales of plankton dynamics and horizontal dispersion regulates plankton blooms and enhances accumulation of photosynthetic biomass in low-nutrient environments. Due to the stimulating effect of the dispersion/dilution process, fine-scale nutrient seeding events may be translated into relatively large ($\sim 10^5$ km²) and long-lived (a few months) phytoplankton blooms.”.

Minor comments: the text could be improved in places for better clarification and consistency, e.g., page 3, line 51, 'this discrepancy is...!', the discrepancy was not clearly explained;

We thank the reviewer for this comment. We replaced “*A major reason for this discrepancy is ...*” with “*A major reason for the limited knowledge about the effect of upscale transfer of plankton variance is..*”.

Figure 1d, 'synthetic particles...!', how were those tracked?

To improve its clarity, we have slightly modified Fig. 1 and its caption. Specifically, the following sentence was added:

“The positions of the particles in time is provided explicitly by the numerical integrator.”.

Page 14 line 217 '...implications of horizontal dilution have barely been explored.', contradicted texts in the introduction, and so on.

As we have noted above, while the “direct cascade” effect of increasing plankton patchiness and enhancing chlorophyll gradients have been subject to a large number of studies, the opposite effect of dispersion and consequent dilution of plankton patches, which are also driven by horizontal stirring, have barely been explored. This is now explicitly mentioned in the manuscript:

References

Abraham, E. R. The generation of plankton patchiness by turbulent stirring. *Nature* **391**, 577-580 (1998).

Dutkiewicz, S., Follows, M. J., Bragg, J. G. Modeling the coupling of ocean ecology and biogeochemistry , *Global Biogeochemical cycles* **23**, doi:10.1029/2008GB003405 (2009).

REVIEWERS' COMMENTS:

Reviewer #3 (Remarks to the Author):

This is an improved version of the manuscript, and the authors attempted to respond to the major points raised earlier; but some points still need to be fixed:

- the phrase "plankton food-web structure" refers to broader context than the nutrient-phytoplankton-zooplankton ecosystem that the authors investigated, so either 'NPZ ecosystem' or 'NPZ dynamics' would be a better choice
- 'larger scales' could be defined more precisely, probably give the range, to distinguish those from fine scales.
- I do not see how "The effect of temperature and light is parametrize by the constant μ ", as μ is simply the growth rate in the model considered, so suggest that the authors delete this, and acknowledge that this simple version of NPZ model does not include those effects explicitly.

Point-by-point response to reviewer's comments (review in blue, reply in black)

Reviewer #3 (Remarks to the Author):

This is an improved version of the manuscript, and the authors attempted to respond to the major points raised earlier; but some points still need to be fixed:

We thank the reviewer for the efforts and for the important comments that helped us making the paper more accurate and clear.

- the phrase "plankton food-web structure" refers to broader context than the nutrient-phytoplankton-zooplankton ecosystem that the authors investigated, so either 'NPZ ecosystem' or 'NPZ dynamics' would be a better choice.

We agree with reviewer that the phrase "plankton food-web structure" is not accurate enough, and we replaced it with (lines 39-40):

"...dynamics of nutrients-phytoplankton-zooplankton ecosystems".

- 'larger scales' could be defined more precisely, probably give the range, to distinguish those from fine scales.

In order to be more precise with the notion of "larger scales", the sentence was re-written as follows (lines 62-63):

" The opposite scenario of fine-scale phytoplankton patches feeding variance back to scales that are larger than the size of mesoscale eddies (typically of the order of 100km) through dispersion is also expected¹."

- I do not see how "The effect of temperature and light is parametrize by the constant μ ", as μ is simply the growth rate in the model considered, so suggest that the authors delete this, and acknowledge that this simple version of NPZ model does not include those effects explicitly.

We agree with the reviewer that the effect of light and temperature is not explicitly represented in the model and we have changed the manuscript accordingly (line 359):
"...and there is no explicit representation of the effect of temperature and light".